# Multipotent Neurotrophic Effects of Hepatocyte Growth Factor in Spinal Cord Injury

**DOI:** 10.3390/ijms20236078

**Published:** 2019-12-02

**Authors:** Kentaro Yamane, Haruo Misawa, Tomoyuki Takigawa, Yoshihiro Ito, Toshifumi Ozaki, Akihiro Matsukawa

**Affiliations:** 1Department of Intelligent Orthopaedic System Development, Okayama University Graduate School of Medicine, Dentistry and Pharmaceutical Sciences, Okayama 700-8558, Japan; woodblocks0311@gmail.com; 2Department of Orthopaedic Surgery, Okayama University Graduate School of Medicine, Dentistry and Pharmaceutical Sciences, Okayama 700-8558, Japan; p36l5a8c@okayama-u.ac.jp (H.M.); pgtx60z9@okayama-u.ac.jp (T.T.); tozaki@md.okayama-u.ac.jp (T.O.); 3Nano Medical Engineering Laboratory, RIKEN, Saitama 351-0198, Japan; y-ito@riken.jp; 4Department of Pathology and Experimental Medicine, Okayama University Graduate School of Medicine, Dentistry and Pharmaceutical Sciences, Okayama 700-8558, Japan

**Keywords:** hepatocyte growth factor, spinal cord injury, neural regeneration

## Abstract

Spinal cord injury (SCI) results in neural tissue loss and so far untreatable functional impairment. In addition, at the initial injury site, inflammation induces secondary damage, and glial scar formation occurs to limit inflammation-mediated tissue damage. Consequently, it obstructs neural regeneration. Many studies have been conducted in the field of SCI; however, no satisfactory treatment has been established to date. Hepatocyte growth factor (HGF) is one of the neurotrophic growth factors and has been listed as a candidate medicine for SCI treatment. The highlighted effects of HGF on neural regeneration are associated with its anti-inflammatory and anti-fibrotic activities. Moreover, HGF exerts positive effects on transplanted stem cell differentiation into neurons. This paper reviews the mechanisms underlying the therapeutic effects of HGF in SCI recovery, and introduces recent advances in the clinical applications of HGF therapy.

## 1. Introduction

Spinal cord injury (SCI) often occurs due to a physical damage that results in a contusion or compression of the spinal cord. It is estimated that over 10 million new SCI cases occur annually worldwide [1,2]. The initial injury destroys the neural tissue at the impact site. Moreover, immune cells are recruited and release cytotoxic factors that can trigger secondary tissue damage. Finally, reactive astrocytes form glial scars to limit inflammation; however, this process inhibits neural tissue regeneration during the chronic stage of SCI. Myelin-associated proteins, including myelin-associated glycoprotein (MAG), Nogo, oligodendrocyte-myelin glycoprotein (OMgp), are expressed by neurons and myelinating oligodendrocytes, and inhibit axonal growth after SCI [3]. These sequential changes and cytotoxic factors hinder spontaneous neural regeneration and lead to irreversible neurological disabilities.

Hepatocyte growth factor (HGF) was first identified as a mitogenic protein for rat hepatocytes in 1984 [4,5]. Since its discovery, extensive studies have revealed the potential therapeutic activities of HGF for the treatment of various diseases of the liver, kidney, or lung. HGF is in the spotlight as one of the neurotrophic growth factors that exerts pleiotropic effects on the central nervous system. Recent studies have revealed that HGF exerts multiple neurotrophic activities in SCI [6,7,8,9]. Our previous study showed promising results for a new SCI combination therapy, using a scaffold and a novel HGF fused to the collagen-binding domain (CBD) derived from fibronectin (CBD-HGF) [7]. To gain further insights into the effects and benefits of HGF on neural tissue regeneration, we review the molecular mechanisms underlying the therapeutic effects of HGF in the spinal cord. Additionally, we focus on recent advances in the clinical applications of HGF therapy in nerve regeneration and recovery after SCI.

## 2. HGF/c-Met Expression within Injury Site after SCI

c-Met is the only HGF receptor known, and it is widely expressed by cells of epithelial and endothelial origin [10,11,12]. HGF/c-Met signaling is essential for HGF to exert therapeutic effects in various cells and tissues. In major organs, such as the liver, kidney, and lung, HGF activity and its interaction with c-Met increase immediately after an injury [13,14,15]. HGF is supplied both from the injured tissues (endogenous reserve), and from other organs, through the blood stream. In the spinal cord, the dynamics of HGF biosynthesis seem to be different. Previous studies have confirmed that HGF and c-Met are produced endogenously in the spinal cord, after SCI in young rats [6,16]. c-Met mRNA was up-regulated at the injury site immediately after SCI, and its expression levels remained elevated in the chronic phase. In contrast, HGF mRNA expression remained at basal levels immediately after SCI, and peaked approximately one or two weeks after SCI. The amount of HGF protein at the injury site increased gradually during the first week after SCI, and persisted at high levels in the chronic phase. Reactive astrocytes surrounding the injury site are one of the candidate cells that could be responsible for HGF secretion [16]. The therapeutic effects of HGF may be limited due to the limited amount of endogenous HGF during the acute phase of SCI. It was suggested that endogenous HGF within the spinal cord of aged mice could be enough to assure therapeutic effects when combined with neural stem cell (NSC) transplantation, in the acute phase of SCI [17]. These reports partially revealed the dynamics of HGF production in animal spinal cord injury; however, the dynamics of HGF and c-Met expression in human spinal cord after SCI remain unknown.

## 3. Neurotrophic Activities of HGF in SCI

### 3.1. Anti-Inflammatory Effects

Inflammation is triggered immediately after primary mechanical damage, and leads to additional tissue damage mediated by infiltrating leukocytes. To decrease secondary damage, it is important to control the intensity of the inflammatory response. HGF regulates the activity of immune cells in various organs. Macrophages are the primary actors of the inflammatory response in SCI [18]. Classically, macrophage polarization has been classified in two categories: inflammatory, M1-like macrophages, and anti-inflammatory, M2-like macrophages. HGF inhibits M1-like macrophages from producing pro-inflammatory cytokines and chemokines, through the inactivation of nuclear factor kappa B (NF-κB) signaling pathway [19] (Figure 1). This effect arises from the enhancement of the heme oxygenase-1 transcriptional pathway and the inactivation of glycogen synthase kinase-3 beta (GSK-3β) pathway [20,21,22]. HGF also inhibits the infiltration of neutrophils through the down-regulation of adhesion molecules [23]. Our previous report demonstrated that the infiltration of macrophages and neutrophils was reduced under the influence of HGF in the acute phase of SCI [7]. The anti-inflammatory activities of HGF contribute to reduce secondary damage and improve functional recovery after SCI.

### 3.2. Anti-Apoptotic Effects

HGF inhibits the caspase-independent pathway, thus suppressing neuron and oligodendrocyte apoptosis in SCI [6]. Previous reports revealed that the inhibition of apoptosis-inducing factor (AIF) translocation and the inactivation of caspase-3 correlated with the anti-apoptotic activity of HGF [6,24]. However, the precise mechanism underlying the anti-apoptotic effect of HGF remains unclear. As some pro-inflammatory cytokines such as tumor necrosis factor-alpha (TNF-α) and interleukin-1 (IL-1) induce cell apoptosis [25,26,27], the anti-apoptotic effects of HGF may also be correlated with its anti-inflammatory activity (Figure 1).

### 3.3. Angiogenic Properties

HGF has also been shown to promote angiogenesis in various organs [28]. Angiogenesis is essential for tissue repair as it provides nutrients, oxygen, and growth factors [29]. After blood vessel destruction due to the initial and secondary damage, new blood vessels expand from existing capillaries around the injury site. Administration of HGF enhances angiogenesis at the injury site [6]. The anti-apoptotic activity on vascular endothelium and the inhibition of endothelial permeability underlay the angiogenic properties of HGF [30,31]. Macrophage polarization from the M1-like to M2-like phenotype is also connected with the regulation of angiogenesis after SCI [32,33]. Both phenotypes are important for new blood vessel formation. HGF decreases the infiltration of M1-like macrophages during the acute phase of SCI, and may affect angiogenesis through its role in macrophage polarization (Figure 1).

### 3.4. Anti-Fibrotic Effects

HGF can suppress fibrosis in various tissues and organs [34,35,36]. The anti-fibrotic activity of HGF arises from the suppression of the transforming growth factor beta (TGF-β) signaling pathway [37,38]. In the acute phase of SCI, reactive astrocytes form glial scars that are essential to attenuate inflammation and limit the damage of surrounding tissues. On the other hand, chronic glial scars inhibit axon regeneration [39]. Previous reports have revealed that HGF decreases TGF-β secretion and chondroitin sulfate proteoglycan (CSPG) production in reactive astrocytes, in vitro [8] (Figure 1). Furthermore, transplantation of mesenchymal stem cells (MSCs) that overexpress HGF in the acute phase of SCI regulates glial scar formation and CSPG deposition around the injury site, and promotes axonal growth into the lesion site beyond glial scars [8]. Therefore, HGF can suppress glial scar formation and limit inflammation-mediated damage through its anti-inflammatory activity.

### 3.5. Neurogenic and Oligogliogenic Effects

Cell transplantation has emerged as a potential treatment for SCI [40]. NSCs can differentiate into all three major cell types of the central nervous system (neurons, astrocytes, and oligodendrocytes), and may be used for cell transplantation therapy in SCI. However, when transplanted, most NSCs differentiate into astrocytes, whereas fewer generate neurons and oligodendrocytes [41,42]. As astrocyte differentiation promotes glial scar formation, this leads to a decrease in the efficiency of NSCs transplantation therapy. Previous reports have demonstrated that HGF promotes neuronal differentiation of grafted NSCs and enhances synapse formation between new neurons and the descending corticospinal fibers [17,43,44] (Figure 1). Other in vitro reports have shown that HGF also promotes oligogliogenesis of embryonic stem cells [45], and the combination of HGF and glial cell-derived neurotrophic factor stimulates bone marrow stromal stem cells (BMSCs) to differentiate into neuron-like cells [46,47]. These effects of HGF on various stem cells can be used to enhance cell-based therapies in SCI recovery.

## 4. Recent Advances towards Clinical Applications in SCI Recovery

### 4.1. Single Modality Approach

Optimization of the routes of drug administration is essential to increase drug efficacy and address clinical medicines. Previous in vivo animal studies tackling the subject of SCI treatment evaluated different HGF therapy approaches, such as the use of recombinant HGF [9], viral vectors [6], grafting of mesenchymal stem cells that over-express HGF [8], or utilization of engineered HGF with a collagen biding domain (CBD-HGF) [7]. Irrespective of the therapeutic approach, local administration of HGF is required as the endogenous HGF supply is limited in the acute phase of SCI. In these animal studies, HGF treatments were started immediately after injury or before injury. Whereas, it takes at minimum a few hours to start the treatments after SCI in clinical practice. The sooner HGF is administrated, the better the benefits of the anti-inflammatory effects can be expected. There is no in vivo animal study investigating the neurotrophic effects of HGF according to its administration time in the acute phase of SCI. In Japan, one phase I/II clinical trial for acute cervical SCI was conducted by Kitamura et al. [48]. In this trial, recombinant HGF was intrathecally injected 72 h after injury, and repeated weekly, during a five-week period. Although the detailed results have not been announced yet, this clinical trial can unveil the therapeutic effects of HGF at the administration time of 72 h after injury. Based on the result of this trial, further studies are required to explore other possible routes of HGF administration and to determine the therapeutic time-window for maximizing its therapeutic effects. Some clinical trials in other neurological diseases such as amyotrophic lateral sclerosis and diabetic peripheral neuropathy may be helpful to predict the adverse effects of HGF treatments [49,50,51]. As HGF is known to promote malignant progression in tumors and resist anti-cancer drugs because of the cell-protective effect of HGF [52], all the trials excluded patients with malignant neoplasm. In relatively short follow-up periods varying between 64 days and 12 months after treatments, there were no serious side effects reported. Thus, knowledge about HGF in SCI is still limited despite the previous valuable studies. Further studies are required in both animal models and clinical trials for clarifying the mechanisms of its effects and possible side effects.

### 4.2. Combination Therapy Approach

The ideal treatment for SCI is a combination therapy using cells, neurotrophic growth factors, and scaffolds [53,54]. Most of the previous research studies and the clinical trial on HGF therapy for SCI recovery focused on monotherapy approaches. Few in vivo animal reports exist on combination therapies using HGF and cells or scaffolds [7,17]; however, none of them applied all three factors (cells, neurotrophic growth factors, and scaffolds). Therefore, little is known aboutxx the interactions between HGF, transplanted cells, and scaffolds. Takano et al. examined the effects of host-derived HGF on transplanted NSCs in mice affected by SCI [17]. HGF improved the survival of transplanted NSCs and enhanced their neuronal differentiation. To our knowledge, this is the only in vivo study investigating the combination therapy with HGF and transplanted cells. Concerning the use of HGF and scaffolds, our previous study revealed that the combination therapy employing CBD-HGF and a gelatin-furfurylamine hydrogel exerts enhanced neurorestorative properties [7]. CBD-HGF has a higher collagen affinity due to the presence of CBD [55], and remains a longer time at the injury site than native HGF. One administration of CBD-HGF enhanced functional recovery after spinal cord compression injury in mice. CBD-HGF alone failed to improve functional recovery after complete transection injury; however, when combined with the gelatin-FA hydrogel, CBD-HGF promoted endogenous repair and recovery. These results suggest that different optimal combination therapies may be required, according to the type of injury. Future studies should unravel the true potential of combination therapies employing HGF, various cell types, and specific scaffolds (Figure 2).

## 5. Summary

HGF exerts multiple effects on neural regeneration. The main mechanism underlying the neurotrophic activity of HGF seems to be the minimization of secondary damage in the acute phase of SCI. HGF also interacts with various cell grafts, such as NSCs and BMSCs; however, these interactions have not yet been fully explored. Combination therapies associating HGF, various cell types, and scaffolds may promote synergistic, increased positive effects on neural regeneration in SCI.

## Figures and Tables

**Figure 1 ijms-20-06078-f001:**
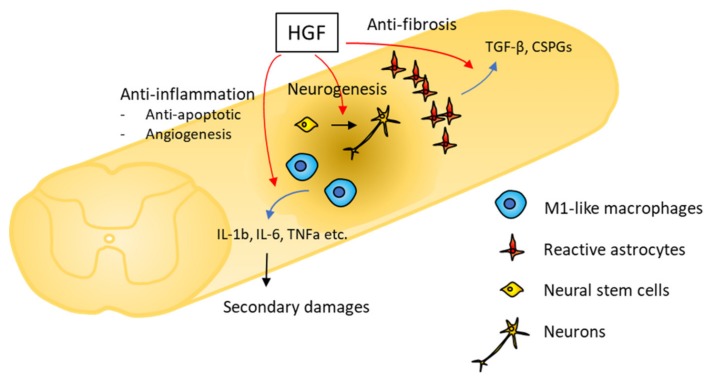
The highlighted benefit of the hepatocyte growth factor (HGF) is related to its anti-inflammatory and anti-fibrotic activities. HGF potentially attenuates the inflammatory cascade to inhibit M1-like macrophages from producing pro-inflammatory cytokines and chemokines. HGF suppresses glial scar formation through the reduction of TGF-β secretion and chondroitin sulfate proteoglycan (CSPG) production in reactive astrocytes. HGF also promotes the neural differentiation of grafted neural stem cells (NSCs).

**Figure 2 ijms-20-06078-f002:**
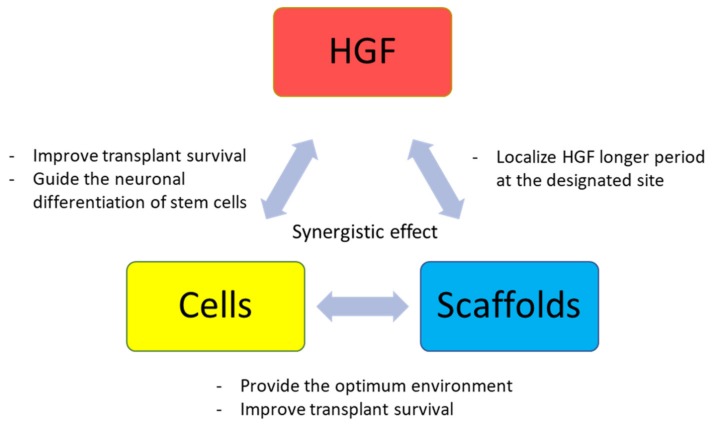
A combination therapy approach employing the hepatocyte growth factor (HGF), various cell types, and scaffolds could be the ideal therapy for efficient recovery after spinal cord injury. HGF can interact with transplanted cells, such as neural stem cells (NSCs) and bone marrow stromal stem cells (BMSCs).

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
