# Peer review of "Multipotent Neurotrophic Effects of Hepatocyte Growth Factor in Spinal Cord Injury"

_ijms, 2019, doi:10.3390/ijms20236078_

Round 1

Reviewer 1 Report

The paper is well written and explains very well all properties of action for HGF in nerural tissue, i.e. SCI in this case.

Author Response

Reviewer #1:

The paper is well written and explains very well all properties of action for HGF in neural tissue, i.e. SCI in this case.

(Our response)

Thank you for your comments. According to other reviewer’s suggestion, we deleted the section for peripheral nerve injury because this review is focused on spinal cord injury. Accordingly, 4 references (old reference No. 48-51) about peripheral nerve injury were deleted and newly added 2 references (new reference No. 51 and 52). Sentences modified in the text are shown in red.

We believe our revised manuscript has been improved after a major revision. We hope that the revised manuscript will be considered acceptable for publication International Journal of Molecular Sciences.

Reviewer 2 Report

The main topic of this review is to summarize the literature on the possible roles of the hepatocyte grow factor (HGF) in neural regeneration after spinal cord injury (SCI).

First, the authors summarize the main events and the main functional consequences of injury to the spinal cord. Then, the authors introduce the HGF and its recent use in SCI. First, they reviewed studies on HGF/c-Met signaling after SCI to move to the neurotrophic activities of HGF in SCI and in peripheral nerve injury. The review terminates with the main possible clinical applications of HGF after SCI.

Strengths and limits

I find the review of potential interest for the spinal cord injury field at large with a logic organization of the different sections that helps the reader to easily follow the work. However, the review requires a more realistic and detailed description of the knowledge on the therapeutic action on HGF. Indeed, the use of HGF after SCI is recent and the studies reviewed by the authors do not clarify what is really known or what is supposed to know (but not investigated with appropriated studies). This aspect is even more relevant when talking about advances toward clinical applications of the HGF (last section of the review) where the authors focus only on the possible routs of administration of HGF and its combination. It is indeed, in this section (that sounds as a repetition of the introduction) that is necessary to give a complete and unbiased overview of the possible use of HGF after SCI with the remaining questions that still need to be answered.  

Major concern

The authors cited about 60 studies in which more than 20 are from one of the authors. If the references about general topics in the review are excluded (e.g. spinal cord injury, inflammation and neural regeneration), the references about the HGF and its effects on SCI are almost exclusively coming from studies of the authors. Furthermore, most of the these citations are recurrent along the review several times in different paragraphs of different sections (for example the references in the introductions to illustrate the effects of the HGF in SCI are repeated in the last sections (advancements toward clinical applications) to again show the different effect on HGF approaches). Since the recurrence of references may be due to a low availability of studies on HGF and SCI, the authors should include some critical points to make the reader aware that:

-knowledge about HGF in SCI is still limited and it needs further experiments in animals models for clarifying the mechanisms of its effects and possible side effects;

-the recent advances toward clinical applications may require not only further investigations for its administration with other factors but actually it still need to be investigated by its own;

Author Response

Reviewer #2:

The main topic of this review is to summarize the literature on the possible roles of the hepatocyte grow factor (HGF) in neural regeneration after spinal cord injury (SCI).

First, the authors summarize the main events and the main functional consequences of injury to the spinal cord. Then, the authors introduce the HGF and its recent use in SCI. First, they reviewed studies on HGF/c-Met signaling after SCI to move to the neurotrophic activities of HGF in SCI and in peripheral nerve injury. The review terminates with the main possible clinical applications of HGF after SCI.

Strengths and limits

I find the review of potential interest for the spinal cord injury field at large with a logic organization of the different sections that helps the reader to easily follow the work. However, the review requires a more realistic and detailed description of the knowledge on the therapeutic action on HGF. Indeed, the use of HGF after SCI is recent and the studies reviewed by the authors do not clarify what is really known or what is supposed to know (but not investigated with appropriated studies). This aspect is even more relevant when talking about advances toward clinical applications of the HGF (last section of the review) where the authors focus only on the possible routs of administration of HGF and its combination. It is indeed, in this section (that sounds as a repetition of the introduction) that is necessary to give a complete and unbiased overview of the possible use of HGF after SCI with the remaining questions that still need to be answered. 

Major concern

The authors cited about 60 studies in which more than 20 are from one of the authors. If the references about general topics in the review are excluded (e.g. spinal cord injury, inflammation and neural regeneration), the references about the HGF and its effects on SCI are almost exclusively coming from studies of the authors. Furthermore, most of the these citations are recurrent along the review several times in different paragraphs of different sections (for example the references in the introductions to illustrate the effects of the HGF in SCI are repeated in the last sections (advancements toward clinical applications) to again show the different effect on HGF approaches). Since the recurrence of references may be due to a low availability of studies on HGF and SCI, the authors should include some critical points to make the reader aware that:

-knowledge about HGF in SCI is still limited and it needs further experiments in animals models for clarifying the mechanisms of its effects and possible side effects;

-the recent advances toward clinical applications may require not only further investigations for its administration with other factors but actually it still need to be investigated by its own;

 (Our response)

We appreciate your thoughtful comments. According to your suggestions, we deleted the section for peripheral nerve injury (previous section 4) and extensively revised “Recent advances towards clinical applications in SCI recovery” (new section 4). The sentences modified in the text are shown in red.

As you pointed out, the studies on HGF and SCI are very limited. There are only 5 in vivo animal studies on HGF and SCI [Reference: 6, 7, 8, 9, 17]. Moreover, considering the difference between in vivo animal studies and clinical use, we proposed some important questions to be answered in future. In clinical practice, the treatments can be started in a few hours after injury. In all animal studies, HGF treatments were started immediately after injury or before injury. We don’t know the effective time-window about HGF treatment after SCI. Clinical trial by Kitamura et al. [Reference: 48] or further studies may help answer this question. The adverse effects of HGF are also important for clinical application. As reference, some clinical trials in other neurological diseases are cited in this manuscript [Reference: 49, 50, 51]. Finally, we added the sentences to make the readers aware the limitation for answering these questions at the end of the section 4.1. [Page.4 line 176-178.].

We believe our revised manuscript has been improved after a major revision. We hope that the revised manuscript will be considered acceptable for publication International Journal of Molecular Sciences.

Reviewer 3 Report

Well written review. Would recommend including a small section on barriers to adoption of HGF toward current therapies, and relevant next steps to utilizing HGF as a treatment for SCI.

Author Response

Reviewer #3:

Well written review. Would recommend including a small section on barriers to adoption of HGF toward current therapies, and relevant next steps to utilizing HGF as a treatment for SCI.

 (Our response)

Thank you for your comments. We agree with your suggestions and revised the manuscript accordingly. We added some descriptions about remaining barriers for clinical use, such as the administration time, and the possible adverse effects.

According to other reviewer’s suggestion, we deleted the section for peripheral nerve injury because this review is focused on spinal cord injury. Accordingly, 4 references (old reference No. 48-51) about peripheral nerve injury were deleted and newly added 2 references (new reference No. 51 and 52). Sentences modified in the text are shown in red.

We believe our revised manuscript has been improved after a major revision. We hope that the revised manuscript will be considered acceptable for publication International Journal of Molecular Sciences.

Round 2

Reviewer 2 Report

The authors have answered my major concern, clearly stating the needs for more experimental data both for the use of HGF in animal models of spinal cord injury and for understanding the translational potentials of this factor in the case of human SCI.